# Circulating miRNA Fingerprint and Endothelial Function in Myocardial Infarction: Comparison at Acute Event and One-Year Follow-Up

**DOI:** 10.3390/cells11111823

**Published:** 2022-06-02

**Authors:** Ana Mompeón, Daniel Pérez-Cremades, Ana Belén Paes, Juan Sanchis, Luis Ortega-Paz, Rut Andrea, Salvatore Brugaletta, Manel Sabate, Susana Novella, Ana Paula Dantas, Carlos Hermenegildo

**Affiliations:** 1Department of Physiology, Faculty of Medicine and Dentistry, University of Valencia, INCLIVA Biomedical Research Institute, Avda. Blasco Ibáñez, 15, 46010 Valencia, Spain; ana.mompeon@northwestern.edu (A.M.); daniel.perez@uv.es (D.P.-C.); anapaesmarti@gmail.com (A.B.P.); carlos.hermenegildo@uv.es (C.H.); 2Cardiology Division, Hospital Clínico Universitario de Valencia (HCUV), INCLIVA Biomedical Research Institute, University of Valencia, Centro de Investigación Biomédica en Red de Enfermedades Cardiovasculares (CIBERCV), Avda. Blasco Ibáñez, 17, 46010 Valencia, Spain; sanchis_juafor@gva.es; 3Institut d’Investigacions Biomèdiques August Pi i Sunyer (IDIBAPS), Carrer del Rosselló, 149, 08036 Barcelona, Spain; lgortega@clinic.cat (L.O.-P.); randrea@clinic.cat (R.A.); sabrugal@clinic.cat (S.B.); masabate@clinic.cat (M.S.); 4Institut Clinic Cardiovascular (ICCV), Hospital Clinic de Barcelona (HCB), Carrer de Villarroel, 170, 08036 Barcelona, Spain

**Keywords:** microRNA profile, serum biomarker, myocardial infarction, endothelial cell, let-7e, angiogenesis

## Abstract

MicroRNAs (miRNA) are major regulators of intercellular communication and key players in the pathophysiology of cardiovascular disease. This study aimed to determine the miRNA fingerprint in a cohort of 53 patients with acute myocardial infarction (AMI) with non-ST-segment elevation (NSTEMI) relative to miRNA expression in healthy controls (*n* = 51). miRNA expression was initially profiled by miRNA array in the serum of patients undergoing cardiac catheterization during NSTEMI (*n* = 8) and 1 year past the event (follow-up, *n* = 8) and validated in the entire cohort. In total, 58 miRNAs were differentially expressed during AMI (*p* < 0.05), while 36 were modified at follow-up (Fisher’s exact test: *p* = 0.0138). Enrichment analyses revealed differential regulation of biological processes by miRNA at each specific time point (AMI vs. follow-up). During AMI, the miRNA profile was associated mainly with processes involved in vascular development. However, 1 year after AMI, changes in miRNA expression were partially related to the regulation of cardiac tissue morphogenesis. Linear correlation analysis of miRNA with serum levels of cytokines and chemokines revealed that let-7g-5p, let-7e-5p, and miR-26a-5p expression was inversely associated with serum levels of pro-inflammatory cytokines TNF-α, and the chemokines MCP-3 and MDC. Transient transfection of human endothelial cells (HUVEC) with let-7e-5p inhibitor or mimic demonstrated a key role for this miRNA in endothelial function regulation in terms of cell adhesion and angiogenesis capacity. HUVEC transfected with let-7e-5p mimic showed a 20% increase in adhesion capacity, whereas transfection with let-7e-5p inhibitor increased the number of tube-like structures. This study pinpoints circulating miRNA expression fingerprint in NSTEMI patients, specific to the acute event and changes at 1-year follow-up. Additionally, given its involvement in modulating endothelial cell function and vascularization, altered let-7e-5p expression may constitute a therapeutic biomarker and target for ischemic heart disease.

## 1. Introduction

Acute coronary syndrome (ACS) stands as the leading cause of death worldwide in men and women (WHO, 2020) and is the clinical manifestation of myocardial ischemia resulting from insufficient blood supply to the myocardium due to reduced coronary artery size. After ACS, a systemic inflammatory response is triggered and then partially suppressed by anti-inflammatory cytokines, promoting cardiac repair [1,2].

Diagnosis of ACS is based on clinical symptoms, electrocardiogram (ECG), and serological markers. ACS includes (1) ST-segment elevation myocardial infarction (STEMI), (2) non-ST-segment elevation myocardial infarction (NSTEMI) [3], both associated with elevated myocardial damage biomarkers such as troponin, and (3) unstable angina, non-related to elevated troponin levels in the circulation [4,5]. However, troponin levels in NSTEMI patients are not persistently high within the first hours of symptoms onset, requiring sequential measurements and hindering early diagnosis [6]. Moreover, despite being an established biomarker for ACS, false-positive troponin elevations are found in a variety of other syndromes, including type 2 NSTEMI and Myocardial Infarction with no Obstructive Coronary Arteries (MINOCA), leading to a need to combine clinical and other biochemical markers, among which miRNAs have lately attracted particular interest [7].

Damaged heart tissue has been reported to cause the release of non-coding RNAs (ncRNAs), including microRNAs (miRNAs), analogous to the release of cardiac troponins [8]. miRNAs are small single-stranded RNAs (~19–23 nt) that function as important post-transcriptional regulators of gene expression. The growing interest in circulating miRNAs lies in their potential use as clinical biomarkers for diagnosis and prognosis and therapeutic targets for multiple diseases. In particular, dysregulation of miRNA biogenesis has been associated with many cardiovascular diseases [8], and several studies suggest their potential use as molecular biomarkers for STEMI diagnosis [9,10,11,12,13]. Increased levels of cardiac-specific and muscle-enriched miRNAs, such as miR-1, miR-133, miR-208, and miR-499, have been found in the bloodstream of AMI patients [14,15,16,17]; nevertheless, given the atypical clinical presentation and higher uncertainty in the diagnosis of NSTEMI patients, analyzing their circulating miRNA profiles might be particularly useful in the search for additional clinical biomarkers.

In addition to the promising use of circulating miRNAs as clinical biomarkers, the presence of stable extracellular miRNAs in the bloodstream suggests that these small molecules may act as active hormone-like mediators in an autocrine, paracrine, or even endocrine manner [11,18]. Intercellular communication involving circulating miRNAs may be essential for endothelial cells as they line the inner surface of blood vessels [19]. Nonetheless, despite promising findings supporting the hypothesis of extracellular miRNAs as Intercellular communicators in the context of the cardiovascular system [20,21,22,23,24] it is still unclear whether they mediate significant biological roles distant from the cell from which they originate.

This study aimed to determine a circulating miRNA expression fingerprint in NSTEMI patients at two different time points: at acute presentation and 1 year after the event (follow-up). We also sought to identify the biological pathways potentially modulated by miRNA during the pathophysiology and course of AMI, which may help develop potential therapeutic biomarkers and targets for ischemic heart disease involved in inflammation and vascularization.

## 2. Materials and Methods

### 2.1. Study Population and Blood Sampling

In our study, peripheral blood samples were obtained from 53 patients admitted to the Hospital Clinic de Barcelona (HCB) or Hospital Clinic Universitari de Valencia (HCUV) for percutaneous coronary intervention (PCI) due to acute myocardial infarction (AMI group, *n* = 53) (Table 1). Inclusion criteria were age <75 years, with first acute myocardial infarction without ST elevation (NSTEMI) and more than one traditional cardiovascular risk factor. Primary exclusion criteria were myocardial infarction with ST-elevation (STEMI) and unsuccessful PCI. Exclusion criteria also included documented previous acute coronary syndrome or stroke, thrombocytopenia, vasculitis or any known immunological disorder, uncontrolled hypertension (systolic or diastolic arterial pressure >180 mmHg or 120, respectively, despite medical therapy), or limited life expectancy (e.g., neoplasms). Then, 1 year after PCI, all patients included in the study were invited for a hospital follow-up visit. Only 38 patients agreed to donate a blood sample for serum and miRNA isolation and had a follow-up in their clinical history (Follow-up group). Samples from volunteers (Control group, *n* = 51) without a history of acute myocardial infarction or chronic coronary disease were included as controls for comparative purposes. Some of the control samples were provided by the Biobank IBSP-CV (PT13/0010/0064) within the Spanish National Biobanks Network and the Valencian Biobanking Network, and other control samples were obtained from volunteers from HCB. Blood samples were collected by venipuncture into SST™ Vacutainer tubes prior to PCI and at 1-year follow-up, as previously described [25]. Experiments were carried out under the approval of the Clinical Research Ethics Committee (CEIC) of the HCB (CEIC2013/8812) and HCUV (CEIC 2014/01/30), in accordance with the principles of the Declaration of Helsinki (64th General 15 Assembly of the World Medical Association, Fortaleza, Brazil, October 2013). Patients were informed of the experimental protocols and provided signed informed consent.

### 2.2. miRNA Isolation and Expression Profile

Total circulating RNA, including small RNA and miRNAs, was isolated from 400 µL aliquot of non-hemolyzed serum, as previously described [25]. Samples obtained from eight control subjects, eight patients prior to PCI, and eight matched follow-up patients were randomly selected from those with the greatest quantity and quality of RNA for miRNA profiling. Equal amounts of miRNA (10 ng) were used for reverse transcription and preamplification using TaqMan™ miRNA Reverse Transcription and preamplification Kits (Applied Biosystems, Thermo Fisher, Waltham, MA, USA). Expression of 754 unique miRNAs was determined by TaqMan™ Human MicroRNA Panels A and B (Applied Biosystems, Thermo Fisher, Waltham, MA, USA). Relative expression of miRNA was calculated according to the ΔΔCt method considering control samples as a reference. Relative quantification was performed using QuantStudio™ Software (Thermo Fisher, Waltham, MA, USA) and expressed as Log2 of fold change.

### 2.3. Functional Enrichment of miRNA Profile

Prediction of biological processes modified by miRNA profile during the AMI event and at 1-year follow-up was made by initially generating a network of miRNA/gene interaction using the miRNet platform (https://www.mirnet.ca/, accessed on 20 May 2020) [26]. miRNA–mRNA interactions were directly downloaded from the webpage and plotted for network visualization on Cytoscape Software V3.8 (San Diego, CA, USA). Gene Ontology and Kyoto Encyclopedia of Genes and Genomes pathway analyses were performed using ClueGO plug-in [27] for visualization and integration of miRNA/genes network with their molecular processes. In silico analysis of miRNA-mediated pathway regulation was performed by using the miRSystem database [28]. A specific network of miRNA and biological processes associated with angiogenesis was constructed to select the miRNAs to be validated across the cohort.

### 2.4. miRNA Validation by Quantitative Real-Time PCR Assay (qRT-PCR)

Expression of individual angiogenesis-associated miRNAs was determined in the entire cohort by qRT-PCR analysis using the 7900HT Fast Real-Time PCR System (Applied Biosystem-Thermo Fisher, Waltham, MA, USA) and a pre-established protocol [25]. cDNA samples were used for amplification by qRT-PCR using TaqMan microRNA Assays (Applied Biosystem-Thermo Fisher, Waltham, MA, USA) of the selected miRNAs and TaqMan 2X Universal PCR Master Mix, No AmpErase UNG (Applied Biosystem-Thermo Fisher, Waltham, MA, USA). miRNAs were chosen for validation in the entire cohort based on significant correlation with biological processes associated with the pathophysiology and outcomes of myocardial infarction. miR-484 expression was used for normalization purposes, as previously described [25]. Each sample was analyzed in duplicate, and expression was calculated according to the 2^−ΔΔCt^ method considering the average cycle threshold (Ct) value of all control samples as a reference. Relative quantification was performed using SDS 2.1 Software (Applied Biosystem-Thermo Fisher, Waltham, MA, USA).

### 2.5. Endothelial Cell Culture

Human umbilical vein endothelial cells (HUVEC, Lonza, Barcelona, Spain) were grown in Medium 199 (Sigma-Aldrich, Madrid, Spain) supplemented with 20% fetal bovine serum (Gibco, Invitrogen, Barcelona, Spain), endothelial cell growth supplement from bovine neural tissue (Sigma-Aldrich, Madrid, Spain), and heparin sodium salt from porcine intestinal mucosa (Sigma-Aldrich, Madrid, Spain). Cells were routinely grown in an incubator at 37 °C with 5% CO_2_.

### 2.6. miRNA Transfection

miRCURY LNA microRNA inhibitor and mimic for let-7e-5p, and microRNA inhibitor and mimic negative control (scrambled) oligonucleotides were purchased from Exiqon. Both oligonucleotides were labeled at the 5′ end with 6-FAM Fluoresceine and transfection efficiency was evaluated by flow cytometry using BD LSRFortessa X-20 (Becton Dickinson, Franklin Lakes, NJ, USA). The optimal transfection conditions for a transfection efficiency of over 80% were determined as follows. HUVEC transfections were performed by using the DharmaFECT 4 Transfection Reagent (Dharmacon, Lafayette, CO, USA) according to the manufacturer’s instructions. Briefly, 1.5 × 10^5^ cells in exponential growth phase were cultured in six-well plates containing 1.8 mL endothelial cell media without antibiotics (Cell Applications, San Diego, CA, USA). A final concentration of 20 nM of let-7e-5p inhibitor or 30 nM of let-7e-5p mimic, respectively, combined with 2 µg/µL DharmaFECT 4 Transfection Reagent in 200 µL Opti-MEM Medium (Gibco-Thermo Fisher, Waltham, MA, USA) was incubated for 20 min at room temperature. The complex was then added to the cells and swirled cautiously over the entire plate surface. After 18 h of incubation, the culture media was replaced by complete endothelial cell media (Cell Applications, San Diego, CA, USA) and the cells were incubated for an additional 24 h. Cells transfected with scrambled-LNA were cultured parallel to the LNA microRNA inhibitor and mimic transfected cells.

### 2.7. Endothelial Cell Adhesion and Proliferation Assays

Adhesion capacity of HUVEC was evaluated on fibronectin (Becton Dickinson) treated dishes (2.5 µg/cm^2^). In brief, 5 × 10^4^ transfected cells were seeded on cell culture dishes with a 2 mm^2^ grid (Nunc, Thermo Fisher Scientific, Madrid, Spain) and incubated at 37 °C and 5% CO_2_. After 30 min, non-adherent cells were washed. A total of 10 random squares were imaged using a Nikon digital sight Ds-QiMc camera and Nikon Eclipse-Ti inverted microscope. Attached cells were counted by using ImageJ 1.49d Software (NIH, Bethesda, MD, USA). Data were expressed as the average number of adhered cells compared to the negative control. Cell proliferation was measured by flow cytometry with propidium iodide (PI) staining (Immunostep, Salamanca, Spain) to quantify DNA content and cell population distribution along the different cell cycle phases. Briefly, transfected cells were starved for 24 h and then stimulated for 18 h with endothelial cell complete culture media (Cell Applications, San Diego, CA, USA). After incubation, cells were detached with a 0.05% trypsin solution (Gibco-Thermo Fisher, Waltham, MA, USA), fixed with 70% ethanol and stained with PI-RNase solution (Immunostep, Salamanca, Spain). Stained cells were analyzed with FACSVerse cytometer (Becton Dickinson, Franklin Lakes, NJ, United States) and ModFit LT Software (Verity Software House, Topsham, ME, USA). Results were shown as the percentage of cells undergoing DNA synthesis plus mitosis (S + G2/M cells) relative to total cell number.

### 2.8. Angiogenesis Assay In Vitro

Transfected endothelial cells were seeded at 1.5 × 10^4^ cells/well on 96-well plates coated with Matrigel^®^ matrix diluted 1:1 with serum-free complete endothelial cell media (Cell Applications) and incubated for 6 h. Capillary-like structure formation was imaged with a Nikon digital sight Ds-QiMc camera and Nikon Eclipse-Ti inverted microscope with a 4× objective (total magnification 40×). Capillary-like structures (meshes) were analyzed and quantified with ImageJ 1.49d Software (NIH, Bethesda, MD, USA). Results were shown as the average number of meshes per image compared to the negative control.

### 2.9. Multiplex Cytokine Assay

Serum levels of a panel of pro-inflammatory cytokines (IL-1β, IL-6, TNF-α, VEGF-A), anti-inflammatory cytokines (IL-10, IL-13) and chemokines (IL-8, MIP-1α, MCP-3, MDC) were evaluated using Milliplex^®^ MAP (Merck, Darmstadt, Germany) following the manufacturer’s instructions. Plates were analyzed on Luminex 100 System (Luminex, Austin, TX, USA). Measurement of circulating cytokine and chemokine levels (pg/mL) was performed using xPONENT 3.1 software (Merk Millipore - Sigma, Burlington, MA, USA).

### 2.10. Data Analysis and Statistics

Data normality was assessed using the Kolmogorov–Smirnov test. Continuous variables were presented as median with interquartile range (IQR). Two-tailed t-test with Welch’s correction were used to analyze differences in participant characteristics. miRNA array results were shown in volcano plots. Changes in miRNA expression in the acute event of AMI or at the follow-up were expressed as fold-change relative to miRNA expression in the control group and compared by Fisher’s exact test. Data were analyzed by multiple *t*-test with Holm-Sidak correction. In the miRNA array, only miRNAs with 2-fold expression change and with *p* < 0.05 were considered significantly modified in each group. miRNA expression by qRT-PCR, cytokine levels and results from in vitro assays were shown in dot plots. The comparison of means across the three independent variables (Control, AMI and Follow-up) was performed by Brown Forsythe and Welch ANOVA, followed by Dunnett’s T3 posthoc multiple comparison analysis. Correlations between miRNAs and cytokines were assessed using Spearman’s rank correlation coefficient. We used an α level of 0.05 to evaluate statistical significance. Data were analyzed using GraphPad Prism vs. 9.0 (GraphPad Software Inc., San Diego, CA, USA).

## 3. Results

Baseline characteristics of the study population and clinical characteristics of controls and AMI patients are presented in Table 1. The control volunteer groups were on average 9 years younger than infarcted patients. Control volunteers also showed a significantly lower incidence of cardiovascular risk factors, including hypertension, hypercholesterolemia, and diabetes than AMI patients. Nonetheless, there were no differences in body mass index or percentage of tobacco smoking, and sex distribution of samples was also similar in the two groups. Study design and workflow are shown in Figure 1. 

### 3.1. Circulating miRNA Expression Profile during AMI and at One-Year Follow-Up

Circulating miRNA profiles of patients with NSTEMI at the acute event (AMI) and 1 year after the event (Follow-up) were interrogated and compared to samples from healthy controls (Control). In each case, RNA was isolated from *n* = 8 subjects. Expression levels of 754 miRNAs in circulation were determined by miRNA array, in which expression of 542 miRNAs was undetectable in the serum of all groups (data not shown). The circulating miRNA profile in AMI patients differed substantially in the acute event compared to follow-up. In total, 58 miRNAs were differentially expressed during acute NSTEMI (*p* < 0.05, vs. Control), while 36 miRNAs were modified at follow-up (Fisher’s exact test, *p* = 0.0138, vs. AMI) (Figure 2A and Appendix A). Volcano plots showed that most differentially expressed miRNAs in patients with AMI were downregulated (91% in acute phase vs. 78% at follow-up; Fisher’s exact test *p* = 0.074). Of these differentially expressed miRNAs, 36 were modified only during the acute event and 14 were altered 1 year after myocardial infarction (Figure 2A). The specific miRNAs altered in the acute phase and follow-up, and those shared across both groups are shown in Figure 2B.

Gene Ontology (GO) enrichment analysis of differentially expressed miRNAs in each group revealed differential regulation of biological processes at each specific time point. When compared to healthy controls, miRNA modified in the acute phase of AMI are mostly those associated with: (1) vascular development, (2) cardiomyocyte differentiation, and (3) cellular response to vascular endothelial growth factor stimulus (Figure 3 and Appendix A). At 1 year after myocardial infarction, however, changes in miRNA expression were partially associated with the regulation of: (1) cardiac muscle tissue development, (2) angiogenesis, and (3) vasculature development (Figure 4 and Appendix A).

### 3.2. Validation of Circulating miRNA Expression in the Study Population

Inadequate formation of vascular networks is a critical process in the pathophysiology of myocardial infarction, contributing both to the acute event and long-term outcomes. In this regard, we selected a group of miRNAs that were modified in the array analysis and associated with biological processes involved in vessel formation (Figure 5A). We performed a comparative analysis of miRNA expression by qRT-PCR in control subjects (*n* = 51), the entire study population of NSTEMI patients (*n* = 53) and those included in follow-up (*n* = 38). Compared to control volunteers, let-7e-5p, let-7g-5p, and miR-26a-5p serum expression were markedly decreased during acute NSTEMI (Figure 5B). Nonetheless, only the miRNA let-7e-5p expression was significantly increased in the follow-up compared to the acute event.

### 3.3. Correlation of Circulating miRNA with Serum Levels of Cytokines and Chemokines

Cytokine and chemokine signaling pathways were among the regulatory networks predicted by the miRSystem database (Appendix A) to be modified by a selected set of miRNAs involved in acute NSTEMI. Serum levels of IL-6 and TNF-α pro-inflammatory cytokines were significantly increased in patients after acute onset of NSTEMI compared to control. Although the significant increase in IL-6 was lost at follow-up, TNF-α levels remained significantly increased 1 year after myocardial infarction (Appendix A). We found no differences in the concentration of the remaining cytokines (Appendix A) or chemokines (Appendix A) studied. IL-13 was excluded from the study as most samples expressed undetectable levels of the cytokine.

We next sought to determine the association of expression of circulating miRNAs in modified NSTEMI with cytokine or chemokine levels involved, plotting the 2^−∆∆Ct^ values of selected miRNA against the concentration of each cytokine and chemokine in the respective sample. A linear correlation analysis including all samples in the study (*n* = 142) revealed anti-inflammatory regulation of cytokines for some miRNAs (Table 2). Let-7g-5p and let-7e-5p were inversely associated with the expression of the pro-inflammatory cytokine TNF-α (Table 2). Moreover, the let-7g-5p expression has a negative correlation with the chemoline MCP-3 (Table 2). Another interesting inverse correlation was also found between miR-26a-5p expression and the chemokine MDC (Table 2).

### 3.4. The Role of Let-7e-5p in Modulating Endothelial Cell Function

Cell-to-cell communication involving extracellular miRNAs is particularly significant for endothelial cells, which show high expression of let-7e-5p [29,30]. In light of this, we performed transient transfection experiments with let-7e-5p inhibitor and mimic to assess the effect of this miRNA expression in endothelial cell function modulation. HUVEC were used as an in vitro model for studying endothelial cell adhesion, proliferation, and angiogenesis. Our study revealed a key role for let-7e-5p in regulating endothelial function in terms of cell adhesion and angiogenesis capacity. In contrast, cell cycle differences were not modified by either transfection with let-7e-5p inhibitor or mimic (Figure 6). Cell proliferation was increased under all stimulated conditions compared to starved cells (Figure 6A). However, the effect of let-7e-5p inhibitor and its mimic resulted in negligible changes in cell population distribution along the different phases of the cell cycle. HUVEC transfected with let-7e-5p mimic showed a 20% increase in adhesion capacity compared to cells transfected with scrambled oligonucleotides, whereas there was no change when let-7e-5p expression was silenced with miRNA inhibitor (Figure 6B). We next used Matrigel™ tube formation assay to evaluate the role of let-7e-5p in the angiogenesis capacity of endothelial cells. With this assay, we found that inhibiting let-7e-5p resulted in a 20% increase in the number of tube-like structures (meshes) by HUVEC (Figure 6C).

## 4. Discussion

Patients with NSTEMI show an atypical clinical presentation and higher uncertainty in their diagnosis. In this regard, determination of miRNA expression profiles could be of particular utility. In this work, we assessed the circulating miRNA profile in patients with NSTEMI at two time points: (1) during an acute event, and (2) at 1-year follow-up, finding a distinct miRNA profile at these two study time points which revealed the temporal miRNAs changes in response to cardiac injury in patients with NSTEMI. Enrichment analysis of the two miRNA profiles correlates with specific biological processes involved, being vascular development and cardiac tissue morphogenesis the most enriched pathways, respectively. These findings reinforce the hypothesis of stage-specific miRNAs that may contribute to AMI pathophysiology.

To validate the miRNAs involved in acute NSTEMI, we selected a set of miRNAs associated with regulation of vascular formation based on our initial screening by TaqMan Arrays. Changes in miRNA expression of miR-26a-5p, let-7e-5p, and miR-let7g-5p at acute NSTEMI event were then validated in serum in a larger population. Circulating miRNA dysregulation at the time of acute NSTEMI supports their potential use as biomarkers for the diagnosis and management of NSTEMI patients. In this vein, growing evidence supports the role of the miR-26 family in cardiovascular diseases [31], although their circulating levels in patients with AMI remain unclear. In contrast to our results, elevated circulating miR-26a-5p expression has been found in the plasma of AMI patients and an experimental model of AMI in mice [32,33]. However, other published studies report miR-26a-5p downregulation in serum and plasma from patients with AMI [34,35], in accordance with our findings. These contradictory results could be explained by differences in blood sampling and methods used for miRNA normalization. On the other hand, let-7e-5p and let-7g-5p belong to the let-7 family of miRNAs, extensively studied in the literature and closely associated with endothelial cell function and inflammation [29,36,37,38,39]. Let-7e-5p expression was confirmed to revert to control levels after one year across the entire study population, reaffirming the potential usefulness of this family as non-invasive biomarkers for early diagnosis of NSTEMI. To our knowledge, only one study has associated let-7e-5p with cardioprotective effects, induced in that case by ischemic post-conditioning in the rat heart [40].

We further analyzed the potential biological function of let-7e-5p during an acute NSTEMI event. Previous studies showed increased let-7e-5p expression in ox-LDL-treated HUVEC, promoting NF-κB activation and translocation into the nucleus by inhibiting IκBβ, a crucial inhibitor of the NF-κB signaling pathway [37]. Studies have reported an increase in inflammatory adhesion molecules and inflammatory cytokines after HUVEC transfection with let-7e-5p mimic, while transfection with let-7e-5p inhibitors induced no significant differences [37]. Nevertheless, another study showed that let-7e-5p acts as an anti-inflammatory mediator by abolishing NF-κB activation in epithelial cells [41]. These results might reflect cell-specific differences, in which the same miRNA could exert opposite effects depending on cell type. Relatively few studies have assessed the role of let-7e-5p in endothelial function. Kong et al. demonstrated that the over-expression of let-7e-5p increases endothelial cell migration and vasculogenesis capacity of human and rat endothelial progenitor cells [42]. However, our findings in HUVEC indicated that let-7e-5p inhibitor and mimic transfection increased angiogenic and adhesion capacity, respectively. These controversial results could partly be explained by the different cell types used. At the same time, stimulation of endothelial cell proliferation is crucial to promote re-endothelialization after ischemia injury, as it is an essential feature in vascular healing and vascular function improvement [43]. However, we detected no change in HUVEC proliferative capacity after transfection with either let-7e-5p inhibitor or mimic. Taken together, these results suggest that let-7e-5p is potentially involved in AMI pathology by regulating endothelial adhesion and angiogenesis.

As described above, different miRNAs have been associated with the pathophysiology of AMI. However, little is known about the crosstalk of miRNA and immune mediators in NSTEMI. We detected significant associations between the expression of NSTEMI-related miRNAs and serum levels of selected cytokines and chemokines involved in AMI. Notable among these, let-7e-5p was inversely associated with pro-inflammatory cytokine TNFα. Accordingly, inflammatory activity has been suggested for let-7e-5p, whose overexpression inhibits TNFα levels in epithelial cells [44] and macrophages [45,46]. In addition, a statistically inverse correlation was also found between let-7g-5p and TNFα and MDC-3, while a negative association was found between miR-26a-5p and MDC expression levels. However, none of these associations have been published to date.

Several limitations should be considered when interpreting these results. Firstly, significant correlations had weak correlation coefficient values (r < 0.3), which might be due to the limited sample size. Secondly, the present study identifies an miRNA-cytokine relationship but does not show causality. The negative associations detected between miRNA expression and cytokine or chemokine levels suggest that miRNAs directly target cytokine or chemokine transcripts; nonetheless, the potential roles of circulating miRNAs and cytokine and chemokine in NSTEMI pathogenesis remain to be elucidated.

Different analysis platforms (microarrays, RNA-seq, qRT-PCR) are now being used for initial screening to quantify miRNA expression and identify circulating miRNA profiles. Among them, qRT-PCR is the most sensitive and commonly used quantification method [47,48], although it presents several challenges, including data normalization [25,49,50]. TaqMan Arrays are based on qRT-PCR methodology, containing the same TaqMan MicroRNA Assays used in qRT-PCR analysis to determine individual miRNA expression. Despite the above, the variability found between TaqMan array and qRT-PCR individual assays in our results could be due to the great difference in sample size. The pre-amplification step performed for miRNA profile analysis by TaqMan Arrays might also potentially bias expression results. Nevertheless, hundreds of qRT-PCR reactions can be performed using the same reaction conditions, and larger RNA input volume can be accommodated by using TaqMan array plates. Consequently, they are better suited for profiling applications in samples with low miRNA-concentrations, such as serum [47,48].

Circulating miRNAs have been pinpointed as promising clinical diagnosis biomarkers in AMI in several studies, and others have proposed that combining miRNA expression with established biomarkers of cardiac injury such as high-sensitive (hs) cardiac troponin I or T (hs-cTnI or hs-cTnT) could contribute to more accurate and efficient identification of AMI [13,17,51,52]. For instance, Oerlemans et al. have demonstrated the enhanced diagnostic value of several cardiomyocyte-enriched miRNAs: miR-1, miR-499 and miR-21, when combined with hs-cTnT levels [17]. Interestingly, combining these three miRNAs results in a significantly higher independent diagnostic value than hs-cTnT alone [17]. Similarly, a meta-analysis showed a comparable independent diagnostic performance between these cardiac-specific and muscle-enriched miRNAs and hs-cTnT commercial immunoassays [51]. Some publications report differences in miRNA expression between patients with NSTEMI and STEMI [13,15,53,54] or unstable angina [17,55,56,57,58], contributing to their differential diagnosis. In addition, several associations between circulating miRNAs and a high risk of adverse cardiovascular events in NSTEMI patients have also been uncovered [59]. For example, circulating miR-28 levels are positively correlated with the Global Registry of Acute Coronary Events (GRACE) score [59], a validated risk score for mortality in patients with NSTEMI [60].

Despite growing evidence over the last decade about the potential clinical application of extracellular miRNAs as diagnostic and prognostic indicators of diseases, their role as intercellular mediators and the biological significance of miRNAs as endocrine signals remains a matter of debate. However, different pieces of data provide evidence supporting a potential “hormone-like behavior” of circulating microRNAs [61,62,63]. Along the same lines, selective sorting mechanisms for miRNA packaging into small extracellular vesicles (sEV) have been identified [64,65,66], supporting the hypothesis of the biological function of sEV-associated miRNAs. In the context of the cardiovascular system, the role of extracellular miRNAs as intercellular communicators between different cell types has been reported [22,23,24]. In this regard, sEV-mediated miRNA transfer between different cell types has been suggested to perform a biological function in cardiovascular diseases and provide a cardioprotective role [20,21,24]. In addition, experimental studies have demonstrated that sEV-mediated miRNA delivery may be incorporated within the atherosclerotic lesion. For instance, Zernecke et al. show the transfer of miR-126 from HUVEC apoptotic bodies giving rise to higher levels of miR-126 and atheroprotective effects in recipient vascular cells [67]. In accordance, differential regulation of circulating miRNAs across the coronary circulation has been demonstrated in ACS patients [68]. Muscle-enriched miRNAs increase, whereas the endothelial miR-126 is picked up during transcoronary passage [68]. Similarly, Fichtlscherer et al. report a vascular-associated miRNA downregulation in patients with coronary artery disease [69]. While several pieces of evidence suggest that miRNAs may indeed participate in cell-cell crosstalk between cardiovascular cells, the biological significance of their function in an endocrine manner remains to be elucidated.

Although our data suggest that the miRNA profile is associated to the acute event of myocardial infarction, since this profile is differently expressed from healthy individuals and modified 1 year after the acute event, we cannot rule out the influence of other factors such as comorbidities, presence of coronary disease (not defined in the control group), or chronic pharmacological treatments on the expression of the analyzed miRNAs. In this regard, we have performed a sub-analysis on the expression of the main miRNA studied (i.e., let-7e-5p) by health status and treatments. In the sub-analysis by health status, the expression of let-7e-5p was determined in AMI/NSTEMI patients in comparison with non-infarcted controls grouped as (1) healthy individuals without diagnosed cardio-vascular risk factors (*n* = 23); (2) non-infarcted patients with at least one diagnosed risk factor (*n* = 28); (3) all non-infarcted controls (healthy + risk factors, *n* = 51). Moreover, to rule out the influence of pharmacological treatments in the expression of let-7e-5p, data in the AMI group were subdivided into patients with no previous exposure to a specific treatment (naïve groups), and those receiving statins, antithrombotic therapy, inhibitors of the renin-angiotensin system, beta-blockers, or diuretics. In the sub analysis by health condition, although we found a trend decrease in let-7e-5p expression in association with cardiovascular outcome (from health to cardiovascular risk to NSTEMI), the levels of circulating let-7e-5p was still markedly reduced in NSTEMI patients when compared to control individuals regardless of the presence of cardiovascular risk factors (Appendix A). Moreover, receiver-operating characteristic (ROC) curve analysis in distinction between NSTEMI and different controls, as wells as, naïve, and treated NSTEMI patients described that NSTEMI variable has the higher discriminatory power in let-7e-5p expression (Appendix A).

Overall, in this study we provide evidence that circulating miRNA profiles are differentially modified at the acute onset of NSTEMI and 1-year follow-up and have identified miRNA expression signatures that could help to find candidate biomarkers, among which let-7e-5p stands out as a promising biomarker for early diagnosis of patients with suspected NSTEMI. Furthermore, given its involvement in modulating endothelial cell function, let-7e-5p might be a potential therapeutic target for ischemic heart disease. Finally, we detected statistically significant correlations between miRNAs involved with acute NSTEMI and inflammatory mediators of AMI, suggesting that deregulation of circulating miRNAs and their association with pro-/anti-inflammatory cytokine and chemokine levels have diagnostic potential in NSTEMI, contributing to NSTEMI pathogenesis.

### Study Limitation

Despite the novelty and relevance of our data, we are aware of the limitations of this study, many of which are still unknown variables in the field. For instance, it is not easy to exclude the influence of factors other than NSTEMI pathogenesis on the regulation of miRNA expression. As a primary effect we have performed an analysis on the expression of the main miRNA studied (i.e., let-7e-5p) in subgroups divided by health status and treatments at the time of inclusion. Even though, our analysis showed no correlation between treatments or healthy status in the control groups with let-7e-5p expression, we are aware that the size of our cohort and the time point in our study may not provide sufficient statistical power to allow to completely rule-out their interference. We are also conscious that the size of our cohort could lead to Type-1 error, albeit our careful statistical analysis. However, the fact that the results involving the expression and effects of let-7e-5p showed an extremely high level of statistical significance, *p* < 0.001, in a small cohort, provides a degree of confidence to support our hypothesis. Our results, however, shed much light on this topic by showing the presence of specific fingerprint in NSTEMI patients. The predictive or therapeutic potential of these mRNAs must be analyzed by our group, and by others, in a larger cohort with more clinical information on patient’s outcomes.

## Figures and Tables

**Figure 1 cells-11-01823-f001:**
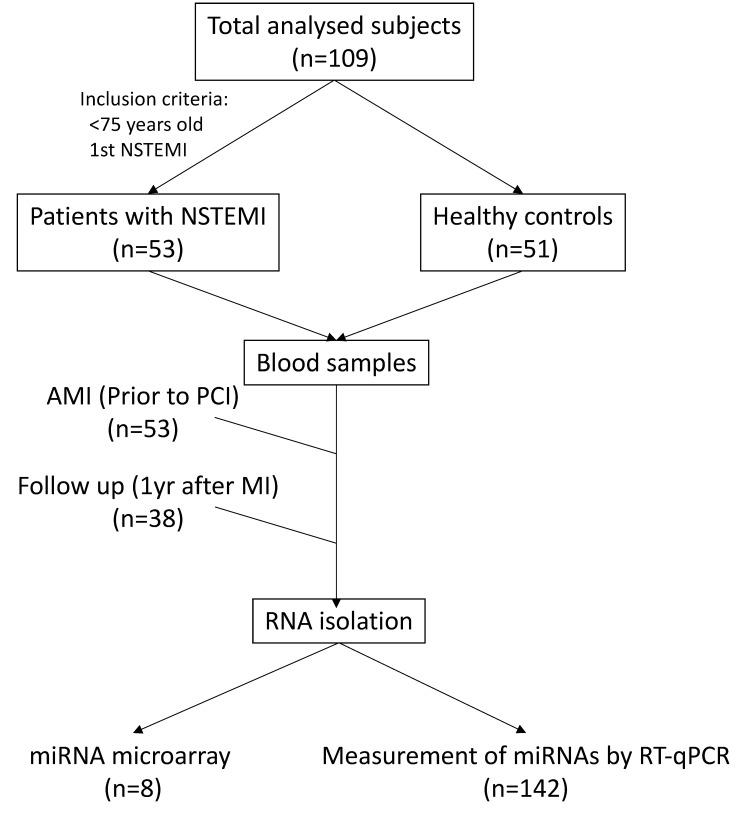
Study design and workflow.

**Figure 2 cells-11-01823-f002:**
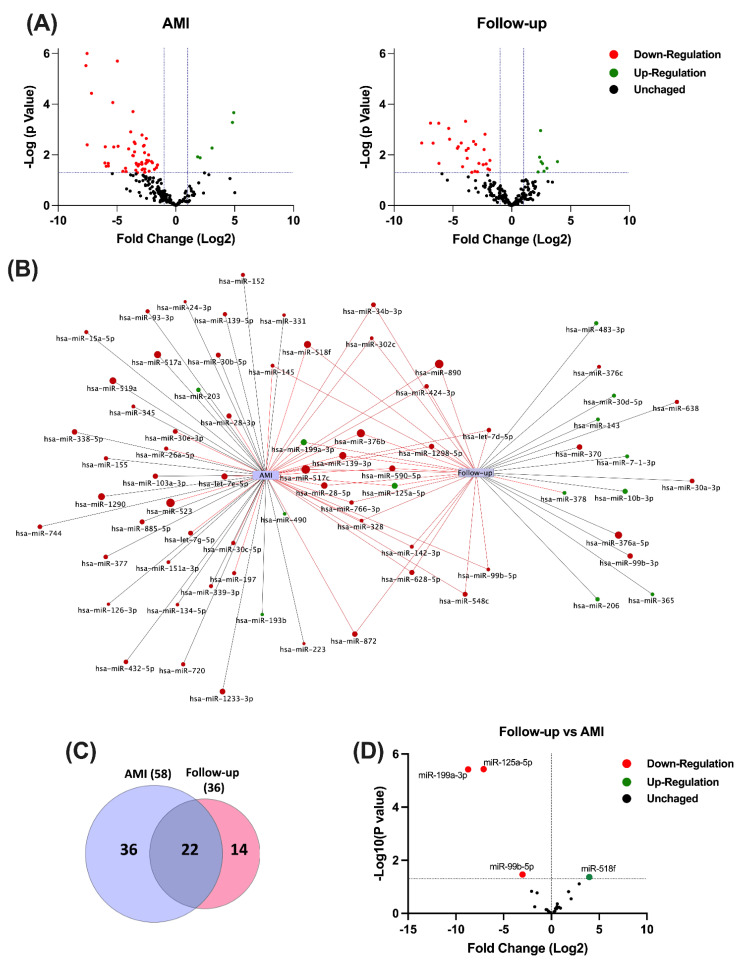
Circulating miRNA expression profile in serum of NSTEMI patients during acute event (AMI) and after 1 year (Follow-up). (**A**) Volcano plots show the values of fold change (Log2, x-axis) by the adjusted *p*-value (−Log10 of *p*-value, y-axis). The colors represent genes or miRNAs that were at least 2-fold downregulated (red) or upregulated (green) in AMI (*n* = 8) and Follow-up (*n* = 8) relative to healthy control samples. (**B**) Network visualization of miRNAs significantly modified in the acute event AMI and at 1-year follow-up (*p* > 0.05). Shown are the miRNA exclusively expressed in each group (black edges) and those shared across both groups (red edges). Size of nodes are associated to the degree of fold change (Log2) of miRNAs that are down-regulated (red nodes) or up-regulated (green nodes). (**C**) Venn diagram and hierarchical clustering representing differentially expressed miRNAs found in AMI and Follow-up groups. (**D**) Volcano plot showing the comparative expression [Fold change (Log2), x-axis] by the adjusted *p*-value (−Log10 of *p*-value, y-axis) of miRNAs that were shared across AMI and Follow-up groups. Relative changes in miRNA expression were analyzed by multiple *t*-test with Holm–Sidak correction.

**Figure 3 cells-11-01823-f003:**
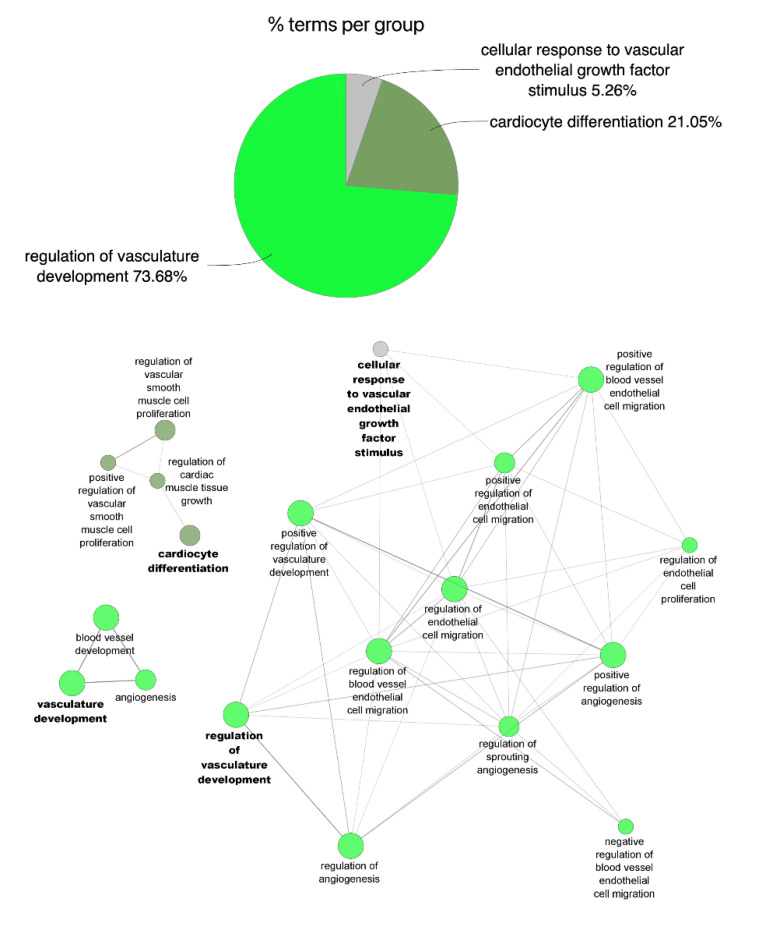
Functional enrichment of miRNA profile and associated biological process in the acute event of acute myocardial infarction (AMI). The pie chart plots the percentage of miRNA-associated functional biological processes that were significantly modified (*p* < 0.05, log2 FC ≥ 1/≤−1) in AMI group. Overrepresented Biological Process Gene Ontology (GO) terms are represented as nodes, and node size represents term enrichment significance by Bonferroni test (*p* < 0.05).

**Figure 4 cells-11-01823-f004:**
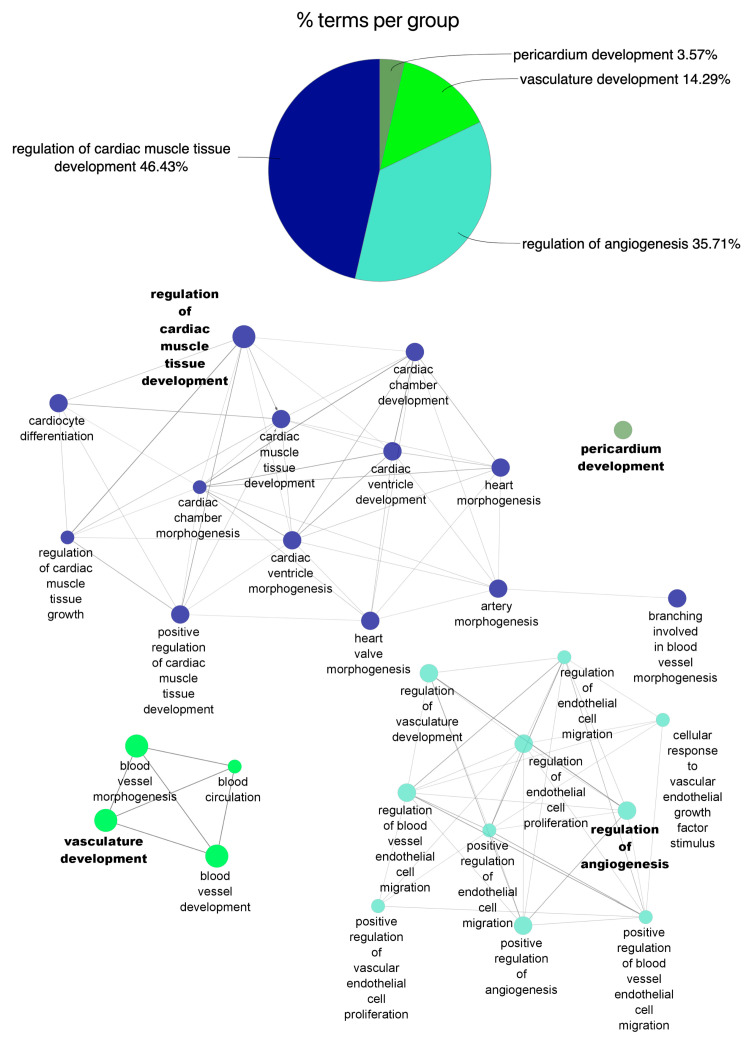
Functional enrichment of miRNA profile and associated biological process after 1 year after acute myocardial infarction (Follow-up). The pie chart plots the percentage of miRNA-associated functional biological processes that were significantly modified (*p* < 0.05, log2 FC ≥ 1/≤−1) in Follow-up group. Overrepresented Biological Process Gene Ontology (GO) terms are represented as nodes, and node size represents term enrichment significance by Bonferroni test (*p* < 0.05).

**Figure 5 cells-11-01823-f005:**
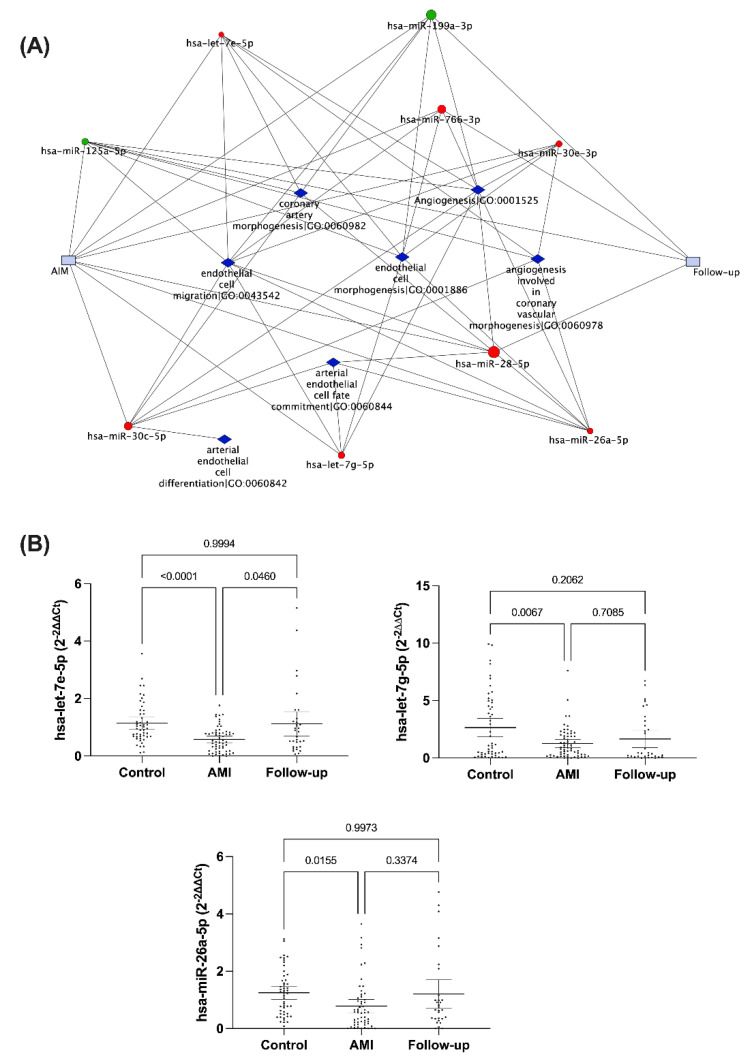
Validation of circulating miRNA expression in the study population. (**A**) Network plot representing association of miRNAs modified in NSTEMI and their association with biological processes involved in vessel formation. (**B**) Comparison of circulating miRNA expression in patients with NSTEMI and control subjects. Data represent the fold change (2^−ΔΔCt^) of the selected miRNAs regarding the acute NSTEMI event, shown as geometric mean with 95% CI. *p*-values were calculated using Brown Forsythe and Welch ANOVA (Control *n* = 51; NSTEMI *n* = 53; Follow-up *n* = 38).

**Figure 6 cells-11-01823-f006:**
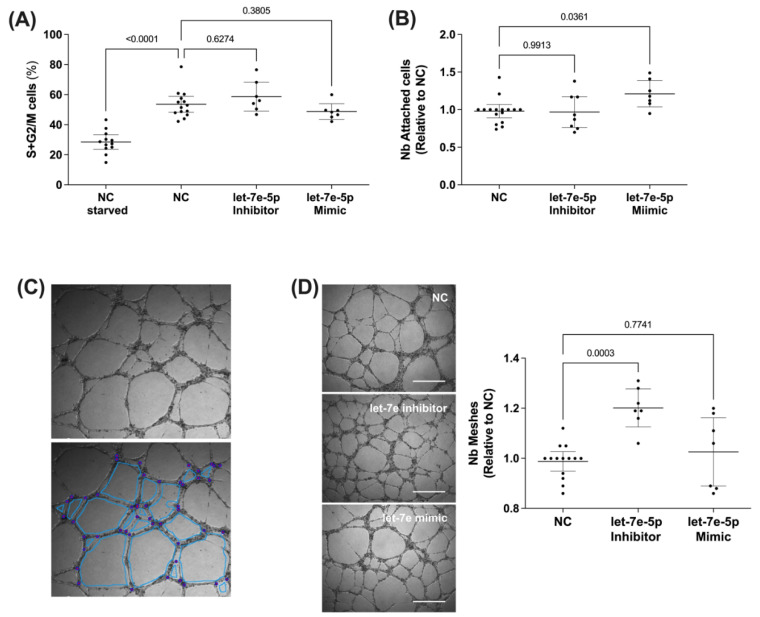
Role of let-7e-5p in endothelial cell proliferation, adhesion, and tube formation. (**A**) Cell proliferation capacity of transfected HUVEC with let-7e-5p inhibitor and let-7e-5p mimic. After transfection for 18 h, cells were starved for 24 h then stimulated with complete endothelial cell media for an additional 18 h. Cells were harvested, stained with propidium iodide solution, and measured by flow cytometry. Results represent the number of cells in proliferation (interphase plus mitosis), expressed as a percentage of total cells. (**B**) After HUVEC transfection with let-7e-5p inhibitor and let-7e-5p mimic for 18 h, adhesion assays were performed as detailed in the Methods section. Data representing the number of adhered cells relative to negative control (NC) are shown as mean ± SEM (*n* = 8). (**C**) Representative image of tube-formation assay (top) and the parameters analyzed (bottom). Blue/red dots represent the number of junctions, and light blue are the meshes formed by endothelial cells. (**D**) Angiogenesis capacity after HUVEC transfection with let-7e-5p inhibitor and let-7e-5p mimic. A total of 50,000 transfected cells were seeded onto Matrigel matrix and incubated for 6 h. Results represent the number of meshes formed compared to negative control (NC), expressed as mean with 95% CI. Representative images are shown on the right (original magnification ×40). Scale bar represents 500 µm. *p*-values were calculated using Brown Forsythe and Welch ANOVA.

**Table 1 cells-11-01823-t001:** Baseline characteristics.

	Control Group (*n* = 51)	AMI Group (*n* = 53)	*p*-Value
Age, median (IQR)	60 (46–64)	65 (56–71)	<0.001
Sex [male, *n* (%)]	30 (58.8)	39 (73.6)	0.147
BMI, median (IQR)	27.2 (24–29)	27.4 (25–30)	0.327
Diabetes mellitus, *n* (%)	2 (3.9)	21 (39.6)	<0.001
Hypertension, *n* (%)	15 (29.4)	45 (84.9)	<0.001
Hypercholesterolemia, *n* (%)	10 (19.6)	31 (58.5)	<0.001
Smoking, *n* (%)	18 (35.3)	19 (35.8)	0.953

Data are shown as a number and percentage (%) or median with interquartile range (IQR), as indicated. BMI—Body Mass Index; AMI—acute myocardial infarction. Continuous variables were compared with unpaired two-tailed *t*-test with Welch’s correction, and categorical variables were compared with Mann-Whitney U test and chi-square test.

**Table 2 cells-11-01823-t002:** Correlations between circulating miRNAs in AMI and serum levels of cytokine and chemokines.

		Cytokines (pg/mL)					Chemokines (pg/mL)			
miRNA		IL-1β	IL-6	TNF-α	VEGF-A	IL-10	IL-8	MIP-1α	MCP-3	MDC
let-7e-5p	r	−0.10	−0.06	**−0.20**	−0.04	0.05	0.003	−0.08	−0.01	−0.03
	*p*	0.2634	0.5544	**0.0265**	0.6290	0.6128	0.9667	0.3717	0.8854	0.7374
let-7g-5p	r	−0.14	−0.03	**−0.23**	−0.14	−0.10	0.06	0.06	**−0.32**	−0.13
	*p*	0.1329	0.7830	**0.0111**	0.1356	0.2943	0.4997	0.5236	**0.0004**	0.1663
miR-26a-5p	r	−0.03	−0.03	−0.16	−0.005	−0.02	0.04	−0.04	−0.15	**−0.22**
	*p*	0.7658	0.7493	0.079	0.9522	0.8668	0.6342	0.6698	0.1001	**0.0134**

The strength of linear association between both variables: miRNA expression fold-change (2^−ΔΔCt^) and cytokine/chemokine serum level (pg/mL) were analyzed in the entire study population (*n* = 142), independently of clinical condition, using the Spearman’s rank correlation coefficient (r). Bold font highlights significant correlations with *p* values.

## Data Availability

The datasets generated for this study are available on request to the corresponding authors.

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
