# Peer review of "Circulating miRNA Fingerprint and Endothelial Function in Myocardial Infarction: Comparison at Acute Event and One-Year Follow-Up"

_cells, 2022, doi:10.3390/cells11111823_

Round 1

Reviewer 1 Report

Please, include the sample size in abstract. Use the appropriate nomenclature in the whole manuscript: -3p or -5p

In some cases, references are too old. For instance, reference #19 is from 2008. There are a number of more recent studies focused on extracellular miRNAs and endothelial cells. Please, note that the role of circulating miRNAs in endocrine signaling is still unclear. Therefore, the statement that support the reference #19 is wrong. In addition, discuss the limitation related to the poor knowledge on circulating miRNAs as hormones.

The control group has not been appropriately selected: Healthy volunteers without a history of acute myocardial infarction or  chronic coronary disease. What about the impact of medications, comorbidities or age on the relationship between NSTEMI and circulating miRNAs? Please, add a group of “diseased controls” or discuss this limitation.

The authors used Mann-Whitney test. Therefore, continuous variables should be described as median [P25;P75]. Why the authors use in some cases non-parametric tests (Mann-Whitney U) and in other parametric tests (Pearson test)?

The impact of type 1 error on the screening results should be discussed.

The use of dot plots is strongly recommended.

Authors should be aware about the impact of heparin on miRNA quantification. The inclusion of an heparinase treatment is recommended. Alternatively, a sensitivity study or the use of heparin as confounding variable should also be considered. In general, the study needs a higher control of confounding factors, in particular those medication that are know to impact on the circulating miRNA levels.

Author Response

Dear Editor and Reviewers,

We are pleased by the detailed review and the recognition of the relevance of our manuscript Cells-1725662. We have addressed all specific issues raised by both reviewers and discussed their suggestions and critiques point by point below each comment. The changes in the revised manuscript are highlighted with track changes in MS Word.

REVIEWER #1

  1. Please, include the sample size in the abstract. Use the appropriate nomenclature in the whole manuscript: -3p or -5p.

We have followed the reviewer’s suggestion and have included de sample size in the abstract and the nomenclature of miRNAs we studied throughout the revised manuscript.

  1. In some cases, references are too old. For instance, reference #19 is from 2008. There are a number of more recent studies focused on extracellular miRNAs and endothelial cells. Please, note that the role of circulating miRNAs in endocrine signaling is still unclear. Therefore, the statement that supports reference #19 is wrong. In addition, discuss the limitation related to the poor knowledge of circulating miRNAs as hormones.

We agree with the reviewer about the references and have updated some of them in the revised manuscript. Moreover, we agree with the reviewer’s comment that despite promising evidence supporting the role of circulating miRNAs as endocrine intercellular communicators in the cardiovascular, it is still unclear whether they mediate significant biological roles distant from the cell from which they originate. We have therefore included this statement in the revised manuscript (page 2). We also highlight this matter of debate in the discussion (paragraph 7).

  1. The control group has not been appropriately selected: Healthy volunteers without a history of acute myocardial infarction or chronic coronary disease. What about the impact of medications, comorbidities, or age on the relationship between NSTEMI and circulating miRNAs? Please, add a group of “diseased controls” or discuss this limitation.

Although we selected a group of healthy volunteers who presented an age and sex distribution that matched as much as possible with infarcted patients, it was not possible to match other factors that may interfere with the expression of miRNAs, such as the presence of coronary artery disease, comorbidities, and treatments. We agree with the reviewer that those factors could interfere and confound with our results. In this regard, we have performed a sub-analysis of let-7e-5p expression by health status and treatments. In the sub-analysis by health status, the expression of let-7e-5p was determined in AMI/NSTEMI patients in comparison with non-infarcted controls grouped as 1) healthy individuals without diagnosed cardiovascular risk factors (n=23); 2) non-infarcted patients with at least one diagnosed risk factor (n=28); 3) all non-infarcted controls (healthy + risk factors, n=51). Moreover, to rule out the influence of pharmacological treatments in the main miRNA studies (i.e., let-7e-5p) patients in the AIM group were subdivided into patients with no previous exposure to a specific treatment (naïve groups), and those receiving statins, antithrombotic therapy, inhibitors of the renin-angiotensin system, beta-blockers, or diuretics. We show this sub-analysis in the new supplemental figure and also we have included a paragraph discussing this issue and our sub-analysis in the “discussion”, as well, as a Study Limitation. We have attached this sub-analysis to the response to reviewer #1.

  1. The authors used Mann-Whitney test. Therefore, continuous variables should be described as median [P25;P75].

We agree with the reviewer that Mann-Whitney test is commonly regarded as a test of population medians and distribution, therefore we have corrected the data on Table 1 of the revised manuscript to express the continuous variables as median and interquartile range (IQR).

  1. Why the authors use in some cases non-parametric tests (Mann-Whitney U) and in other parametric tests (Pearson test)?

We agree with the reviewer that a non-parametric correlation test is more appropriate for this study. We, therefore, have re-analyzed our data on miRNA and inflammatory markers correlation using Spearman’s rank correlation coefficient (r), and have modified the results accordingly in Table 2, along with the results and discussion.

  1. The impact of type 1 error on the screening results should be discussed.

We agree with the reviewer that the size of our cohort could lead to Type-1 error, albeit our careful statistical analysis. However, the fact that the results involving the expression and effects of let-7e-5p showed an extremely high level of statistical significance p < 0.001 in a small cohort, provides a degree of confidence to support our hypothesis. We, however, have added this comment to the study limitation.

  1. The use of dot plots is strongly recommended.

We have followed the reviewer’s recommendation and have modified the graphics to dot plots. Thus, there are new figures 2, 5 and 6.

  1. Authors should be aware about the impact of heparin on miRNA quantification. The inclusion of an heparinase treatment is recommended. Alternatively, a sensitivity study or the use of heparin as confounding variable should also be considered. In general, the study needs a higher control of confounding factors, in particular those medication that are known to impact on the circulating miRNA levels.

We completely agree with the reviewer that Heparin has an important impact on miRNA quantitation. Having that in mind, we have performed a preliminary study to determine the role of blood sample type on miRNA expression (Sci Rep. 2020 Mar 25;10(1):5373. doi: 10.1038/s41598-020-61507-z) and help us in the decision making on of the sampling process for this study. Our published work suggested that serum may be a more proper sample type in circulating miRNA studies in cardiovascular disease when considering its higher sensitivity and lower variability.

We’d like to thank again to reviewers for both their time and thorough and critical evaluation of our manuscript. We have addressed the reviewers’ recommendations and hope the manuscript has gained in clarity and quality.

Reviewer 2 Report

This interesting, well written manuscript by Monpeón et al. provides evidences that circulating miRNA profiles are differentially modified in serum samples of patients undergoing cardiac catheterization after AMI with NSTEMI, compared to both healthy controls and one-year follow-up samples. The author selected a group of miRNAs associated with biological processes involved in vessel formation, let-7g-5p, let-7e-5p, and miR-26a-5p. They found that the expression of these miRNAs was inversely associated with serum levels of pro-inflammatory cytokines IL-1β, TNF-α, and the chemokine MDC, nevertheless the correlation coefficient values are very weak, as discussed. They also highlighted a role of  let-7e-5p  in endothelial cells adhesion and in improving angiogenic ability of HUVEC on Matrigel.

The study was carefully conducted with appropriate controls; results are well presented.

I have only some minor comments:

-The author show in Fig.2 the differentially expressed miRNAs found in AMI and follow-up. It could be interesting to describe how the modulation of the 22 miRNA shared across both groups, change from acute to follow-up group. Is the color code used in Fig. 2B the same used in Fig. 2A?

-Let-7e-5p inhibition and over-expression in Huvec. The authors should show the levels of Let-7e-5p measured after inhibition and over-expression for all the experiments (proliferation, adhesion, capillary like structure formation) and  should explain in material and methods how they calculate and express the number of attached cells and the number of meshes in Fig 6.  In order to improve the comprehension of these experiments, the authors may discuss the significance of the results obtained, for readers not expert of angiogenesis. Is the increase in adhesion capacity observed after transfection with Let-7e-5p mimic in agreement with the increased angiogenesis observed after transfection with Let-7e-5p inhibitor? How do you explain that inhibition of let-7e-5p shows no effect on cell adhesion?

-Check the exponents along the text and put them in apex  (e.g. 104 instead of 104).

Line 117: “essentially” is not appropriate. If you changed anything in the protocol, please specify what.

Line 394: Please specify that in Kong et al is the over-expression of Let-7e-5p which increase angiogenesis of epc.

Author Response

Dear Editor and Reviewers,

We are pleased by the detailed review and the recognition of the relevance of our manuscript Cells-1725662. We have addressed all specific issues raised by both reviewers and discussed their suggestions and critiques point by point below each comment. The changes in the revised manuscript are highlighted with track changes in MS Word.

REVIEWER #2

  1. The author shows in Fig.2 the differentially expressed miRNAs found in AMI and follow-up. It could be interesting to describe how the modulation of the 22-miRNA shared across both groups, change from acute to follow-up group. Is the color code used in Fig. 2B the same used in Fig. 2A?

We agree with the reviewer on the importance of demonstrating the comparative expression of miRNAs that are shared between AIM and follow-up. We have now updated the Fig 2 with this new information and have included the proper legend to explain the color codes.

  1. Let-7e-5p inhibition and over-expression in HUVEC. The authors should show the levels of Let-7e-5p measured after inhibition and over-expression for all the experiments (proliferation, adhesion, capillary-like structure formation) […]

We agree with the reviewer that had expressed the amount of let-7e-5p would be a piece of important information. Nevertheless, there is a limitation in this field, and we cannot precisely measure miRNA mimetic or anti-miR levels in cells, either because the paired anti-miR+miRNA is rapidly degrading or because we may have false-positive results by measuring miRNA mimetic with endogenous miRNA. In this regard, we have rather focused on their functional effects compared to a random sequence of miRNA (scramble miRNA).

  1. […] and should explain in material and methods how they calculate and express the number of attached cells and the number of meshes in Fig 6. In order to improve the comprehension of these experiments, the authors may discuss the significance of the results obtained, for readers not expert of angiogenesis.

We have followed the reviewer’s suggestion and have explained in better detail the methodology and calculation of HUVEC experiments. We have also included a figure explaining the parameters we are measuring in the tube-formation assay (angiogenesis).

  1. Is the increase in adhesion capacity observed after transfection with Let-7e-5p mimic in agreement with the increased angiogenesis observed after transfection with Let-7e-5p inhibitor? How do you explain that inhibition of let-7e-5p shows no effect on cell adhesion?

In angiogenesis, endothelial cell (EC) adherence and growth are the earlier steps in the angiogenic cascade. However, they are not the only ones and their association with angiogenesis is not straightforward. For angiogenesis to occur efficiently during vessel formation, EC mobilization must be a coordinated event with a precise hierarchy comprising three different phenotypes of leader and follower cells. The tube formation assay will determine the capacity of one of those cells to assume the role of leader with specific and directed migratory behavior, and long filopodia extensions that can connect with other angiogenic sprouts. Each one of these steps is controlled by different gene expressions, and likely miRNAs are controlling those steps at different points in the cascade. Therefore, it is not surprising that we see tube-formation being regulated by let-7e-5p, while this miRNA has no effect on adhesion and growth.

  1. Check the exponents along the text and put them in apex (e.g. 104 instead of 104).

We have corrected the exponents along the revised manuscript

  1. Line 117: “essentially” is not appropriate. If you changed anything in the protocol, please specify what.

We have followed the reviewer’s recommendation and have excluded the word “essentially” from the Methods

  1. Line 394: Please specify that in Kong et al is the over-expression of Let-7e-5p which increase angiogenesis of EPC.

We have made the proper correction in the revised manuscript.

We’d like to thank again to reviewers for both their time and thorough and critical evaluation of our manuscript. We have addressed the reviewers’ recommendations and hope the manuscript has gained in clarity and quality. 

Round 2

Reviewer 1 Report

The authors have addressed my concerns.